# A New Hyper-Heuristic Multi-Objective Optimisation Approach Based on MOEA/D Framework

**DOI:** 10.3390/biomimetics8070521

**Published:** 2023-11-02

**Authors:** Jiayi Han, Shinya Watanabe

**Affiliations:** Muroran institute of technology, Muroran 050-0000, Japan; 20096009@mmm.muroran-it.ac.jp

**Keywords:** evolutionary multi-objective optimization (EMO), MOEA/D, hyper-heuristic approach, CMA-ES, IDE, operator switching mechanism, efficiency inspection

## Abstract

A multi-objective evolutionary algorithm based on decomposition (MOEA/D) serves as a robust framework for addressing multi-objective optimization problems (MOPs). However, it is widely recognized that the applicability of a fixed offspring-generating strategy in MOEA/D can be limited, despite its foundation in the MOEA/D methodology. Consequently, hybrid algorithms have gained popularity in recent years. This study proposes a novel hyper-heuristic approach that integrates the estimation of distribution (ED) and crossover (CX) strategies into the MOEA/D framework based on the view of successful replacement rate (SSR) and attempts to explain the potential reasons for the advantages of hybrid algorithms. The proposed approach dynamically switches from the differential evolution (DE) operator to the covariance matrix adaptation evolution strategy (CMA-ES) operator. Simultaneously, certain subproblems in the neighbourhood denoted as B(i) employ the Improved Differential Evolution (IDE) operator to generate new individuals for balancing the high evaluation costs associated with CMA-ES. Numerical experiments unequivocally demonstrate that the suggested approach offers distinct advantages when applied to a three-objective test suite. These experiments also validate a significant enhancement in the efficiency (SRR) of the DE operator within this context. The perspectives and experimental findings, with a focus on the Success Rate Ratio (SRR), have the potential to provide valuable insights and inspire further research in related domains.

## 1. Introduction

A multi-objective optimization model can be defined as follows [1]: (1)$$\min\;F\left(X\right)={\left[f_{1}\left(x\right),f_{2}\left(x\right),\ldots ,f_{1m}\left(x\right)\right]}^{T}\;s.t.g_{i}\left(x\right)\leq 0,i=1,2,\ldots ,p$$
where $f_{i}(x),\{i=1,2,...,m\}(m\geq 2)$ is the objective function;$\;g_{i}(x)$ is the constraint function; $x={\left\{x_{1},...,x_{n}\right\}}^{T}$ is dimensional design variables, and $X=\{x|x\in R^{n},g_{i}\;(x)\leq 0,i=1,2,...,p\}$ is the feasible region of the above formula. The design variable $x={\left\{x_{1},...,x_{n}\right\}}^{T}$ is a certain set of vectors corresponding to a point on $R^{n}$ which is a n-dimensional Euclidean design space; f(x) corresponds to a point on $R^{m}$ which is a m-dimensional Euclidean objective space. In other words, the objective function f(x) represents a mapping as follows:
(2)$$f:R^{n}\to R^{m}$$

The multi-objective evolutionary algorithm based on decomposition (MOEA/D, Zhang and Li, 2007) [2] is a state-of-the-art algorithm that provides a framework for solving multi-objective optimisation problems (MOPs, Coello and Lamont, 2004; Deb, K., 2011) [3,4]. Figure 1 is used to demonstrate the basic concepts of MOEA/D. In the MOEA/D, a set of evenly weighted vectors in the objective space is used to decompose one MOP into multiple subproblems. Each subproblem is treated as a scalar optimisation problem, where a scalarisation function (e.g., the weighted-sum or Tchebycheff approach proposed by Miettinen (1999)) [5] and weight vectors are utilised to coordinate the relationships among the objective functions. Therefore, it is reasonable to utilise genetic evolution algorithms originally designed to solve single-objective optimisation problems (SOPs), especially crossover (CX)-based algorithms such as the differential evolution (DE) and its variants [6,7,8,9], to generate new solutions.

Hansen et al., (2003) acknowledged that the evolution strategy with covariance matrix adaptation (CMA-ES) [10] was an effective algorithm based on the estimation of distribution (ED) (Springer Science & Business Media, 2001) [11] that has been widely applied in various domains. CMA-ES enhances the search process by estimating a more promising region utilising the distribution information of the current population. Although CMA-ES was originally utilised for single-objective optimisation, it can be applied to solve MOPs, as demonstrated by Igel et al., (2007) and Zapotecas-Mart’ınez et al., (2015) [12,13].

As a further comparison between the CX and ED strategies, it was observed that CX was efficient and effective for a global search but tended to become stagnant and was not well suited for problems with strong variable dependence (non-separable problems). Conversely, ED was effective for problems with dependent variables but incurred a huge evaluation cost.

DE and its variants use crossover (CX) and mutation strategies to generate new solutions. In particular, CX is recognised as an efficient approach for creating promising individuals by combining information from two or more existing individuals (parents).

Generally, new individuals (children) are compared to one of their parents, and the children replace their parents if they demonstrate better evaluation values. This is referred to as a successful replacement scenario. Similar to but different from the feasibility ratio (FR) [14], the overall successful replacement rate (SRR) is expressed as follows:(3)$$SRR=\frac{successful\;replacementtimes}{\max\;evaluationtimes}$$

The results of numerical experiments revealed that the SRR in pure MOEA/D-DE exhibited a low level of efficiency for the Walking Fish Group (WFG) [15] test suite, as listed in Table 1. That is, a significant portion of the evaluation cost did not contribute to the final solution. Also, there is an inherent problem when using ED strategies especially CMA-ES to solve MOPs that significantly increases the evaluation cost. This is because many sampling points are required to obtain the probability distribution in the objective space. Based on the above analysis, a hybrid approach was proposed, called MOEA/D-HH, for which Table 1 presents the average SRRs in two-, three-, and five-objective WFG problems. The numerical experiment was initialised with 300 subproblems and 30 variables (1000 subproblems and 32 variables in WFG_5D problems). The iterations were repeated 21 times, aiming to dynamically switch between different generating operators based on the search situation within the MOEA/D framework, with the core concept of ‘recycling- redistribute’.

Specifically, when a CX operator in a subproblem fails to generate new nondominated individuals for several generations, this indicates that CX cannot be expected to discover a new nondominated individual within a finite number of generations. At this time, MOEA/D-HH switches from the CX to the CMA-ES operator for recycling inefficient evaluation costs.

According to its characteristics, the CMA-ES operator has a high probability of estimating the correct search direction for a sub-problem if there are sufficient sampling points. The evaluation cost of evaluating these sampling points comes from the redistribution of the evaluation cost occupied by CX. After switching to CMA-ES, CMA-ES is continued if a new individual can be nondominated. However, CMA-ES switches back to CX if it fails to update the individual to save evaluation costs. The sampling points generated by the CMA-ES are based on the distribution of the current subproblem and its neighbourhood. Even if the sampling points are not nondominated, they still contain useful information regarding the evolutionary process.

To make the most of the evaluation cost of the CMA-ES, the operators of some of the other subproblems in are switched to the improved differential evolution (IDE, Tang et al., 2014) [16] operator when the operators of subproblem are switched to CMA-ES. The IDE is recognised as a powerful algorithm for solving SOPs, but it also requires a higher evaluation cost to solve MOPs. The sampling points generated by the CMA-ES are sufficient for the IDE operator, so there is no need for additional evaluation costs. The IDE reuses the sampling points generated by CMA-ES, which dilutes the high evaluation cost of CMA-ES, thus achieving the purpose of mitigate the evaluation cost.

More details about the CMA-ES and IDE are introduced in Section 2. The workflows of the proposed approach, MOEA/D-HH, and necessary modifications to the CMA-ES and IDE in this framework are discussed in Section 3. The results of the numerical experiments are presented to demonstrate the effectiveness of the proposed approach in Section 4.

This paper aims to present an effective method of combining CX and EX and demonstrate its usefulness through numerical experiments. The proposed approach is designed to synergistically harness the inherent advantages of CX and ED.

The main contributions of this paper are follows:(1)Concept of re-emphasising the necessity for a hybrid algorithm: Different from the previous hybrid algorithms that combine CX and ED strategies, we dynamically ‘recycle’ the inefficient evaluation cost occupied by CX and ‘redistribute’ it to ED from the point of view of successful replacement rate (SRR).(2)Reuse of evaluation costs: Although various evolutionary algorithms based on CX or ED have been proposed in existing studies, most of them focus only on non-dominant individuals. Based on the underlying logic of the ED strategy, those dominated solutions are also considered containing information relevant to evolution. Therefore, in the proposed algorithm, dominated solutions generated by the ED strategy are reused as archive populations, which contributes to new ideas for maintaining diversity and balancing the high evaluation cost of ED strategies in hybrid algorithms.(3)Framework adaptation: In this study, we propose an operator switching mechanism depend on the Efficiency Inspection. Different from the switching mechanism in other hybrid algorithms, our proposal is tailor-made, which fully considers the characteristics of the neighbourhood in the MOEA/D framework.

## 2. Related Works

The basic concepts of the CMA-ES operator and IDE algorithm are discussed in this section. CMA-ES and IDE were originally designed and are commonly used to solve SOPs. However, the necessary modifications were made to incorporate CMA-ES and IDE into the MOEA/D framework to solve MOPS, which will be explained in the following subsections.

### 2.1. CMA-ES

The CMA-ES is known to be an effective evolutionary algorithm for single-objective problems. In CMA-ES, a population of new search points is generated by sampling a multivariate normal distribution. The basic equation for sampling the search points in generation g+1 is expressed as follows (Hansen, 2006) [17]:(4)$$x_{k}^{\left(g+1\right)}\sim N\left(m^{\left(g\right)},\;{\left(\sigma^{\left(g\right)}\right)}^{2}C^{\left(g\right)}\right)\;\mathrm{for}\;k=1,\ldots ,\lambda$$
where ~ denotes the same distribution on the left and right side, $x_{k}^{\left(g+1\right)}\in R^{n}$ denotes the kth offspring (search point) from generation g+1, $m^{\left(g\right)}\in R^{n}$ denotes the mean value of the search distribution at generation g, $C^{\left(g\right)}\in R^{n\times n}$ denotes the covariance matrix at generation g, and λ≥2 for the population size, sample size, and number of offspring. $N\left(m^{\left(g\right)},\;{\left(\sigma^{\left(g\right)}\right)}^{2}C^{\left(g\right)}\right)\sim m^{\left(g\right)}+\sigma^{\left(g\right)}N\left(0,C^{\left(g\right)}\right)\sim m^{\left(g\right)}+\sigma^{\left(g\right)}B^{\left(g\right)}D^{\left(g\right)}N(0,I)$ is the multi-variate normal search distribution.

Calculating $m^{\left(g+1\right)}$, $C^{\left(g+1\right)}$, and $\sigma^{\left(g+1\right)}$ for the next generation g+1 is required to define the complete iteration step. The equations for updating the parameters for distribution are expressed as the following:(5)$$m^{\left(g+1\right)}=\sum_{i=1}^{u}\omega_{i}x_{i:|X|}^{g}$$
(6)$$p_{\sigma }^{\left(g+1\right)}=\left(1-c_{\sigma }\right)p_{\sigma }^{\left(g\right)}+\sqrt{c_{\sigma }(2-c_{\sigma })\mu_{W}}{C^{\left(g\right)}}^{-\frac{1}{2}}\sum_{i=1}^{u}\omega_{i}\frac{x_{i:|X|}^{g}-m^{\left(g\right)}}{\sigma^{\left(g\right)}}$$
(7)$$\sigma^{\left(g+1\right)}=\sigma^{\left(g\right)}\times \exp\left(\min\left(1,\frac{c_{\sigma }}{d_{\sigma }}\left(\frac{\left|\left|p_{\sigma }^{\left(g+1\right)}\right|\right|}{E\left|\left|N\left(0,I\right)\right|\right|}-1\right)\right)\right)$$
(8)$$p_{c}^{\left(g+1\right)}=\left(1-c_{c}\right)p_{c}^{\left(g\right)}+\sqrt{c_{c}(2-c_{c})\mu_{W}}\frac{m^{\left(g+1\right)}-m^{\left(g\right)}}{\sigma^{\left(g\right)}}$$
(9)$$C^{\left(g+1\right)}=\left(1-c_{1}-c_{\mu }+c_{c}\right)C^{\left(g\right)}+c_{1}p_{c}^{\left(g\right)}{\left(p_{c}^{\left(g\right)}\right)}^{T}+c_{\mu }\sum_{i=1}^{\mu }\omega_{i}\frac{x_{i:|X|}^{g}-m^{\left(g\right)}}{\sigma^{\left(g\right)}}{\left(\frac{x_{i:|X|}^{g}-m^{\left(g\right)}}{\sigma^{\left(g\right)}}\right)}^{T}$$
(10)$$\lambda =4+3\left\lfloor \ln n\right\rfloor$$
(11)$$\mu =\left\lfloor \frac{\lambda }{2}\right\rfloor$$
(12)$$\mu_{W}={\left(\sum_{i=0}^{\mu }\left({\omega_{i}}^{2}\right)\right)}^{-1}$$
(13)$$c_{\sigma }=\frac{\mu_{W}+2}{n+\mu_{W}+3}$$
(14)$$d_{\sigma }=1+c_{\sigma }+2max(0,\sqrt{\frac{\mu_{W}-1}{n+1}-1})$$
(15)$$c_{c}=\frac{4}{n+4}$$
(16)$$c_{1}=\frac{2+min(1,\lambda /6)}{{\left(n+1.3\right)}^{2}\mu_{W}}$$
(17)$$c_{\mu }=min(1-c_{1},\sqrt{\frac{2(\mu_{W}-2+{\mu_{W}}^{-1})}{{\left(n+2\right)}^{2}+\mu_{W}}})$$
where n is the number of design variables.

MOEA/D-HH was optimised to introduce CMA-ES into the MOEA/D framework. First, it was assumed that the current generation g was the last generation that used DE to provisionally generate a new individual. In generation g+1, the current sub-problem i uses CMA-ES to generate new individuals. MOEA/D-HH uses the neighbourhood of the current subproblem i to initialise the parameters of CMA-ES, setting $X^{g}={B(i)}^{g}$ at the end of generation g. Subsequently, λ sampling points are generated and evaluated in generation g+1, where a provisional archive population ${A(i)}^{\left(g+1\right)}$ is set up to store the sampling points. It is worth noting that even if a new sampling cannot be found that dominates the current individual, $x_{i}^{\left(g+1\right)}$. This implies that the information contained in ${A(i)}^{\left(g+1\right)}$ can still contribute to finding the direction of evolution. Therefore, $X^{\left(g+1+j\right)}={B(i)}^{\left(g+j\right)}\cup {A(i)}^{\left(g+j\right)}$ was set where $j\in Z_{+}$.

### 2.2. IDE

The original DE strategy can be summarised as follows:(18)$$DE/rand/1:\;v_{i,g}=x_{r0,g}+F_{i}\cdot (x_{r1,g}-x_{r2,g})$$
where $v_{i,g}$ represents the mutation vector. The indices r0, r1, and r2 in the MOEA/D framework are distinct integers uniformly chosen from ${B(i)}^{g}$. However, $x_{r1,g}$ and $x_{r2,g}$ are randomly selected. Therefore, the difference vector $x_{r1,g}-x_{r2,g}$ will occasionally appear in the opposite direction of evolution. IDE is utilised to divide the parent solution into a superior group (SG) and inferior group (IG) using numerical sorting in the objective space to overcome this problem. This strategy is expressed as follows:(19)$$DE/base/differ/cross:v_{i,g}=x_{i,g}+F_{i}\cdot (x_{sg0,g}-x_{ig1,g})$$
where $x_{sg0,g}$ is selected from the SG and $x_{ig1,g}$ is selected from the IG. The difference vector $x_{sg0,g}-x_{ig1,g}$ ensures that the correct direction of evolution is not reversed. 

However, there are some awkward situations for directly setting the parent solution as ${B(i)}^{g}$ when using the IDE strategy to solve MOPs. On one hand, the selections of $X_{sg0,g}$ and $X_{ig1,g}$ are based on numerical sorting. For the diversity of new individuals, the size of ${B(i)}^{g}$ cannot be too small. On the other hand, according to the characteristics of MOEA/D, a large ${B(i)}^{g}$ leads to lower relevance between individuals in ${B(i)}^{g}$. An ideal situation is that there are enough individuals related to the corresponding subproblem, but obviously, evaluating additional individuals to meet this requirement is not efficient. 

Fortunately, the CMA-ES strategy provides an ideal scenario for IDE. ${A(i)}^{\left(g+1\right)}$ provides sufficient individuals to act as the parent solution. At the same time, the sampling points in ${A(i)}^{\left(g+1\right)}$ are closely related to each other. Therefore, as assumed above, the current subproblem i uses the CMA-ES strategy to generate new individuals in generation g+1. In addition, some of the subproblems in B(i) will use the IDE strategy to generate new individuals. The subproblems that use IDE will share ${B(i)}^{\left(g+j\right)}\cup {A(i)}^{\left(g+j\right)}$ as their parent solution.

## 3. The Proposed Algorithm MOEA/D-HH

This study introduces a hyper-heuristic approach called MOEA/D-HH that combines CX and ED strategies within the MOEA/D framework to generate new solutions. The most important aspect of the MOEA/D-HH approach is the incorporation of an adaptive switching mechanism for generator operations. Additionally, the challenge of applying CMA-ES and IDE strategies to solve MOPs in MOEA/D-HH is addressed because CMA-ES and IDE were originally designed for SOPs.

An overview of the MOEA/D-HH flow and the details of important mechanisms are discussed in the following sections.

### 3.1. MOEA/D-HH Framework

The MOEA/D framework can be easily explained using a simple flowchart, as shown in Figure 2. The framework is divided into three main parts: initialisation, reproduction, and updating. These parts are represented by dotted rectangles of different colours. Classical algorithms based on this framework utilise genetic operators to generate an offspring population (new individuals) during the reproduction phase. Subsequently, the newly generated individuals are used to update the current population. The reproduction and update processes continue to iterate until the stopping criteria are satisfied.

Most approaches, such as MOEA/D-DE, utilise a single fixed operator throughout the search process to generate new individuals. Conversely, MOEA/D-HH, which is built upon the MOEA/D framework and shares the same hyper-parameter initialisation method as MOEA/D-DE, introduces the integration of multiple distinct operators for generating offspring. Therefore, an operator selector was necessary during the reproduction phase of MOEA/D-HH. This selector not only determined the mating population but also chose the appropriate generating operator based on the current search condition. An efficiency inspection was performed on the subproblem after the regular update phase to evaluate the current search condition. The results of this inspection served as the criteria for switching between CX- and ED-based operators. This switching mechanism involved two components: a selector operator and efficiency inspection. Further details of this mechanism are presented in Section 3.4. The core process of MOEA/D-HH is illustrated in Figure 3.

### 3.2. ED Based Operator (CMA-ES) in MOEA/D-HH

In MOEA/D-HH, different sub-problems may utilise different operators to generate new solutions. Initially, all the subproblems employ the DE1 operator, as specified in Equation (18). However, a decline in DE operator effectiveness is indicated if it encounters difficulties in identifying the appropriate evolutionary direction for several generations as the search progresses. In such cases, MOEA/D-HH switches the operator from DE1 to CMA-ES. The CMA-ES operator is known for its ability to identify correct search directions by utilising distribution information. Therefore, is expected to be more effective in generating new non-dominated individuals. The parameters of CMA-ES will be initialised assuming that the operator of subproblem i is switched to CMA-ES at generation g, as shown in Table 2.

At the same time, MOEA/D-HH will create a provisional archives population ${A(i)}^{g}=\emptyset$ to store new individuals that are subsequently generated. After the initialisation, the procedure for CMA-ES in MOEA/D-HH is shown as Algorithm 1:
**Algorithm 1:** CMA-ES in MOEA/D-HH Framework**Input:**$\;\mathrm{Setting}\;\mathrm{solution}\;\mathrm{to}\;X={B(i)}^{g}\cup {A(i)}^{g}$.**Step 1:** Sorting X using the fitness value of the scalarising function.**Step 2:**Updating distribution parameters using sorted X.**Step 3:** Generating new solutions:$\;\;\mathrm{New}\;\mathrm{solutions}\;\mathrm{set}\;Y=Ø.\;\mathrm{For}\;i=0,1,\ldots ,U_{count}\times \lambda ,\;\mathrm{do}:$$Y\leftarrow {Y\cup y}_{i}=m+\sigma C\frac{z}{\left||z|\right|},\;\mathrm{where}\;z\sim N(0,I)$.**Step 4:**$\;\mathrm{Repairing}:\;\mathrm{if}\;\mathrm{an}\;\mathrm{element}\;\mathrm{of}\;y_{i}\in Y$ is out of the boundary, its element value is reset to the boundary.**Step 5:**$\;\mathrm{Storing}:\;{A(i)}^{g+1}\leftarrow {A(i)}^{g}\cup Y$**Output:** $y_{best}$, which is the individual with best fitness value in set Y.

The method in Step 3 is equivalent to Equation (4). Specifically, $U_{count}=1,2,\ldots ,U_{max}\in Z$ represents the number of unsuccessful replacements. If $y_{best}$ cannot replace the current best ${x_{i}}^{g}$, then $U_{count}+=1$. 

The CMA-ES strategy generates several new individuals based on the dominance distribution. Obviously, $y_{best}$ can be generated as long as enough individuals are generated that are not dominated by ${x_{i}}^{g}$. However, considering the practical evaluation cost, the volume of set Y cannot be expanded without limit. Therefore, in this study, $U_{max}=3$.

### 3.3. CX Based Operator (IDE) in MOEA/D-HH

Based on the characteristics of the MOEA/D framework, subproblems within the neighbourhood have similar solutions. Moreover, set Y (similar to A(i)) is strongly dependent on B(i), indicating that the individuals in Y are influenced by the current subproblem i and the entire neighbourhood. In addition, archived individuals improve the diversity of the population. 

In MOEA/D-HH, the operator of other subproblems i‘∈[i−λ2, i)∪(i, i+λ2] within the neighbourhood will switch to the IDE operator after the operator of subproblem i switches to CMA-ES, where λ described in Equation (10) is the size of CMA-ES offspring. The IDE procedure in MOEA/D-HH is shown as Algorithm2:
**Algorithm 2:** IDE in MOEA/D-HH Framework**Input:** Setting solution to X=B(i‘)g∪A(i)g.**Step 1:** Sorting and Partitioning: **Step 1.1:** Sorting X using fitness value of scalarising function. **Step 1.2:** Superior group S← top 30% of X.**Step 2:**
Generation of mutation vector.
$\;\mathrm{Randomly}\;\mathrm{select}\;{X_{S}}^{g}\;\mathrm{and}\;{X_{I}}^{g}\;\mathrm{from}\;\mathrm{groups}\;S\;\mathrm{and}\;I,\;\mathrm{respectively}.$$\;\mathrm{Generate}\;\mathrm{new}\;\mathrm{mutation}\;\mathrm{vector}\;\mathrm{as}\;{V_{i‘}}^{g}={X_{i‘}}^{g}+F\cdot ({X_{S}}^{g}-{X_{I}}^{g})$.**Step 3:** Crossover as:$u_{i‘,j}^{g}=\left\{\begin{array}{c} v_{i‘,j}^{g},\;\;if\;(rand\left(0,1\right)\leq CR,\;\;or\;j=j_{rand})\\ x_{i‘,j}^{g} \end{array}\right.$$\;\mathrm{where}\;u_{i‘,j}^{g}$ denotes the trial vector, F denotes the mutation factor, and CR denotes the crossover probability.**Step 4:**$\;\mathrm{Updating}:\;\mathrm{If}\;fit(U_{i‘}^{g})\leq fit\left(X_{i‘}^{g}\right)$$,\;\mathrm{then}\;X_{i‘}^{g+1}=U_{i‘}^{g}$.**Output:** $X_{i‘}^{g+1}$.

### 3.4. Operator Switching

As described in Section 3.1, the operator switching mechanism in MOEA/D-HH consists of two components: selection and efficiency inspection. MOEA/D-HH initialises an empty list $L_{index}=Ø$ to store the indices of these subproblems, which facilitates the tracking of subproblems using CMA-ES. The subproblems execute the CMA-ES strategy if the current subproblem i is present in $L_{index}$. Additionally, the current subproblem i will use the IDE strategy for generating new individuals, if subproblem i is not included in $L_{index}$ and there is a subproblem $i‘\in [i-\frac{\lambda }{2},\;i)\cup (i,\;i+\frac{\lambda }{2}]$ in $L_{index}$, as discussed in Section 3.3. When neither of the above conditions is satisfied, the subproblems execute the DE1 strategy. The selection part of the operator-switching mechanism is summarised in Figure 4.

Efficiency Inspection is the other component of the operator-switching mechanism that is responsible for managing the members in $L_{index}$. The fundamental concept is illustrated in Figure 5. The criterion for identifying inefficiency is when a newly generated individual fails to outperform the current best solution, and this situation persists for a specified number of generations. This specified number of generations is considered a threshold. In our study, we established thresholds of five and three generations for the CX and ED strategies, respectively.

The threshold for the ED strategy was the same as the setting of $U_{count}$ that was proposed in Section 3.2. This was because the evaluation cost of the ED strategy was much higher than that of CX. The criterion during the efficiency inspection of a subproblem using CMA-ES is whether the threshold is exceeded. If the threshold is exceeded, the index of the subproblem is removed from $L_{index}$ and its provisional archive population is cleared simultaneously. Unlike the ED strategy, the CX strategy has a lower evaluation cost that allows for higher tolerance for inefficient situations. 

In numerical experiments, it is common to observe stagnation in the search progress of a certain subproblem for several generations, whereas other subproblems in its neighbourhood continue to update non-dominated individuals with new solutions during the same time period. Therefore, the focus is on the subproblem itself and all other subproblems in its sub-neighbourhood that may exceed the inefficiency threshold when inspecting the efficiency of a subproblem using DE1 or IDE. In this study, the size of the sub-neighbourhood was set to parameter λ.

## 4. Experiments

Experiments were conducted using the WFG test suite (Huband et al., 2005) [15] to assess the effectiveness of the proposed MOEA/D-HH and compare its performance with that of the original MOEA/D-DE. Furthermore, a reference version called MOEA/D-HHF was introduced that utilised fixed operators to examine the efficacy of the switching mechanism in MOEA/D-HH.

In MOEA/D-HHF, the index of a subproblem is represented by i and the size of the neighbourhood is denoted by T. The ith subproblem employs the CMA-ES strategy to generate new individuals only when i%T==0. Subproblem $i‘\in [i-\frac{\lambda }{2},\;i)\cup (i,\;i+\frac{\lambda }{2}]$ uses the IDE strategy, while the remaining subproblems utilise the DE1 operator.

In this experiment, 21 trials were conducted for each approach and the average values were calculated. The penalty-based boundary intersection (PBI) [18,19] scalarising function and WFG test suite were employed as the set of problem instances. The performances of these approaches were evaluated using the inverted generational distance plus (IGD+) metric (Ishibuchi et al., 2015) [20].

The details of the problem instances used in the experiment, measurement method using the IGD+ metric, parameter settings, and analysis of the results obtained from the experiments are provided in the following sections.

### 4.1. Instances and Measuring Methods

The WFG test suite was utilised as a problem instance in these experiments, which has been widely used and offers flexibility in adjusting the number of objectives and decision variables as needed. Test functions and a true Pareto front for two-, three-, and five-objective problems (referred to as WFG_2D, WFG_3D, and WFG_5D, respectively) were generated following the methodology outlined by Huband et al., (2006) [21].

The position and distance parameters were set to k=2 and l=n−k, respectively, for the WFG_2D problems, where n represents the number of variables. The position parameter was set to k=4 for WFG_3D and WFG_5D.

IGD+ was employed as a performance indicator to evaluate the performance of the approaches. The IGD+ metric measures the distance between the obtained solutions and true Pareto front. A lower IGD+ value indicates that the solutions are closer to the true Pareto front, implying better performance.

### 4.2. Parameter Settings

The parameters used for CMA-ES in MOEA/D-HH are described in Section 2.1 and listed in Table 2. The parameters used for the MOEA/D framework are listed in Table 3. The neighbourhood size T in the MOEA/D framework is generally set to be less than 10% of the population size N. This is because having a large neighbourhood can result in a loss of necessary similarity between subproblems. However, the CMA-ES algorithm requires a certain number of individuals in the neighbourhood, and there is a hidden condition regarding the number of offspring, which is expressed as follows:(20)$$T\geq \lambda ,\;\mathrm{where}\;\lambda =4+3\left\lfloor \ln n\right\rfloor$$

Therefore, MOEA/D-HH cannot be applied when the parameters are set to N=150, T=15, and n=100. The conditions for the CMA-ES were not met in this case.

For the WFG problems, the position parameter k must be a multiple of M−1. Additionally, the distance parameter l=n−k must be divisible by k. Consequently, the number of design variables was set as n=32 for the WFG_5D problems, which was the closest number to 30 and satisfied the aforementioned conditions.

The parameter for the PBI scalarising function was set to θ=5. The mutation factor F and crossover probability CR were set to 0.5 and 0.9, respectively, for the DE1 and IDE operators.

### 4.3. WFG_2D Experiments

Numerical experiments were conducted on the WFG_2D problem using five sets of hyper-parameters. The average IGD+ values obtained from these experiments are presented in Table 4, where the parameters N and n represent the number of subproblems and variables, respectively. The DE1 column represents the results of the original MOEA/D-DE, which served as a baseline control and was generated using the *jMetalpy library* (Benítez-Hidalgo et al., 2019) [22]. Columns HH and HHF represent the results for MOEA/D-HH and MOEA/D-HHF, respectively. The results indicated that the original MOEA/D-DE outperformed MOEA/D-HH and MOEA/D-HHF in most cases. However, MOEA/D-HH demonstrated better performance in WFG5 (N=300,n=30,50,100), WFG6 (N=150,n=50; N=300,n=30,50,100), and WFG9 (N=300,n=100). However, MOEA/D-HHF, which served as a control to evaluate the effectiveness of the switching mechanism, performed better than MOEA/D-HH for WFG2 (N=150,n=30,50), WFG6 (N=150,n=30), and WFG9 (N=150,n=30;N=300,n=30). These results provided preliminary evidence of the effectiveness of the adaptive switching mechanism. 

The evolutionary trajectory was analysed, and selected results are presented in Figure 6, Figure 7 and Figure 8. The x coordinate denotes the progress of iterations, whereas the y coordinate represents the average IGD+ values. These figures demonstrate that MOEA/D-HHF exhibited faster convergence during the initial stages of iteration, despite the significant differences between the results of the HHF and the other two algorithms. 

The impacts of MOEA/D-HH on the SRR is presented in Table 5 and Table 6. The bold entries in these tables compare the SRR values of MOEA/D-DE with those of MOEA/D-HH. The rightmost columns provide the specific SRR and picked rate (PR) of each operator in MOEA/D-HH. In these tables, it is a fundamental requirement that the sum of the PR in each row equals one.

From these tables, particularly Table 6, it is clear that the HH-DE(PR) values were remarkably high, while the SRR values of MOEA/D-DE were superior to those of MOEA/D-HH. These high values indicate that MOEA/D-DE performed exceptionally well in the experimental environment.

Consequently, MOEA/D-HH adaptively selected a higher proportion of DE1 operators. This observation is consistent with the MOEA/D-DE and MOEA/D-HH curves shown in the figures.

### 4.4. WFG_3D Experiments

Numerical experiments were conducted on the WFG_3D problems using three different sets of hyper-parameters, and the average IGD+ values are listed in Table 7. In contrast to the results observed for the WFG2_D problems, the original MOEA/D-DE only outperformed MOEA/D-HH in certain instances, such as WFG1 (n=30, 50, and 100), WFG3 (n=30), and WFG9 (n=30).

The performance of MOEA/D-HHF was inferior to that of MOEA/D-HH, highlighting the effectiveness of the operator-switching mechanism. Selected results for the evolutionary trajectory are shown in Figure 9, Figure 10, Figure 11, Figure 12 and Figure 13. MOEA/D-HH did not exhibit superior performance compared to that of MOEA/D-DE in WFG1, WFG3, and WFG9 with n=30, as shown in Figure 9, Figure 10 and Figure 11, respectively. However, the differences in their performances were exceedingly small.

In contrast, MOEA/D-HH exhibited a significantly better performance than those of MOEA/D-DE and MOEA/D-HHF for WFG4, as shown in Figure 12. In addition, MOEA/D-HH demonstrated superior performance on WFG8, as shown in Figure 13. WFG8 can be considered as a challenging problem owing to the variations in the distance-related parameter values among the different Pareto optimal solutions.

Figure 14 displays the solutions obtained through optimization for the WFG8 problem. In this figure, the red points represent our proposed MOEA/D-HH algorithm, while the blue and green points correspond to the reference MOEA/D-DE1 and MOEA/D-HH, respectively. Upon careful observation, it becomes apparent that, regardless of the projection on any dimension, the red points consistently cluster closer to the origin than the points in other colours. In Pareto optimization terms, this indicates that the evolutionary results of MOEA/D-HH have a greater dominance over the solutions obtained by other methods. This result aligns with the outcomes of IGD+.

The SRR and PR results are listed in Table 8. Similar to the WFG_2D problems, MOEA/D-HH generally exhibited lower SRR values than those of MOEA/D-DE. However, the HH-DE-SRR values were significantly higher in the MOEA/D-HH group. The dominance relationship between the solutions was less likely to be generated and the solutions were more likely to be non-dominated as the number of objectives increased. This indicates that searching is very difficult in multi-objective problems.

The random-based CX strategy incurs a high evaluation cost to explore a vast search space in multi-objective problems. The distribution-based ED operator incurs a higher evaluation cost for generating new individuals than the CX operator. However, the individuals generated by the ED strategy have a higher likelihood of being in the correct search direction (promising regions). This is because the individuals generated by ED take advantage of the approximate gradient information.

### 4.5. WFG_5D Experiments

Numerical experiments were conducted on the WFG_5D problem using a single set of hyper-parameters, and the average IGD+ values are presented in Table 9. The experimental results exhibited interesting patterns as the number of objectives increased. Although MOEA/D-DE performed well in the WFG1 problem, MOEA/D-HHF demonstrated its superiority for the first time in the WFG2, WFG5, and WFG6 problems.

To provide further insight, selected results of the evolutionary trajectory are presented in Figure 15, Figure 16 and Figure 17. MOEA/D-HHF achieved the best overall performance with a faster convergence rate during the early iterations, as shown in Figure 15. However, the convergence rate of MOEA/D-HH exhibited a significant improvement when the number of evaluations exceeded 40,000, achieving a final result very close to that of MOEA/D-HHF.

MOEA/D-HH presented substantial advancements in the search process, particularly during the middle stage, as shown in Figure 16. Specifically, the evolutionary trajectory of MOEA/D-HH was significantly different from those of the other methods after the middle stage of the search. This behaviour can be attributed to the favourable compatibility between the switching mechanism and characteristics of this problem.

The improvements and evolutionary trajectory observed in MOEA/D-HH clearly indicate its superior performance compared to that of MOEA/D-DE.

The same is true for the difficult WFG8 problem, as shown in Figure 17. The SRR and PR results are listed in Table 10. In contrast to previous results, the SRR values of MOEA/D-HH were generally higher than those of MOEA/D-DE for WFG_5D problems, and HH-DE(PR) achieved the highest values compared to those of the 2D and 3D problems.

HH-IDE(PR) and HH-CMA(PR) exhibited relatively lower values. Conversely, HH-IDE-SRR and HH-CMA-SRR were even higher than HH-DE-SRR in some cases, which was not observed in the 2D and 3D problems. These results are consistent with the expectations for MOEA/D-HH.

## 5. Conclusions

Numerous studies have highlighted the limitations of relying on only one single offspring-generation strategy in optimization algorithms, which has led to the increasing popularity of hybrid evolutionary algorithms. In our paper, we introduce the MOEA/D-HH method, and we aim to elucidate the underlying factors contributing to the advantages of hybrid algorithms. Within this proposed algorithm, we have devised an adaptive operator switching mechanism rooted in the concept of operator efficiency inspection, specifically focusing on the successful replacement rate (SRR). This mechanism takes into consideration the specific characteristics of the MOEA/D framework and strives to balance the evaluation costs between the CX and ED strategy. Empirical support for the effectiveness of this switching mechanism is provided through experimental results. 

Furthermore, the experimental results indicate that operators (DE1 and IDE) based on the CX strategy take a mainstream in the hybrid algorithm MOEA/D-HH (they are selected with a higher probability by the switching mechanism). Simultaneously, from the perspective of the SRR, the inclusion of non-mainstream operators (the ED operator) significantly enhances the search efficiency of DE1. The significant improvement in the mainstream strategy (or operator) at the SRR level can directly impact the overall performance of the algorithm, even when the overall SRR of the hybrid algorithm does not exhibit substantial fluctuations. This phenomenon is particularly evident when MOEA/D-HH is applied to the 3-objective test suite. We hope that our research can provide fresh insights for related studies.

However, we must point out that, since our research is based on the MOEA/D framework, we give priority to comparing the proposed algorithm with the original MOEA/D under several different combinations of hyperparameters. The comparison of MOEA/D-HH with other dominant- and indicator-based algorithms is not mentioned in the paper. This will also be the focus of our next work. 

Furthermore, the experimental results revealed certain limitations. One limitation was the fluctuation in the PR of the IDE operator under different experimental conditions. This variability highlighted the need for further investigation and analysis to better comprehend the factors that influence PR and to address the underlying causes of these fluctuations. Additionally, this experiment only considered the PBI scalarisation function. It is necessary to incorporate and explore alternative scalarisation functions in future studies to obtain a more comprehensive understanding of the effects of different scalarisation functions. These limitations should be addressed in future work to refine and enhance the proposed approach.

## Figures and Tables

**Figure 1 biomimetics-08-00521-f001:**
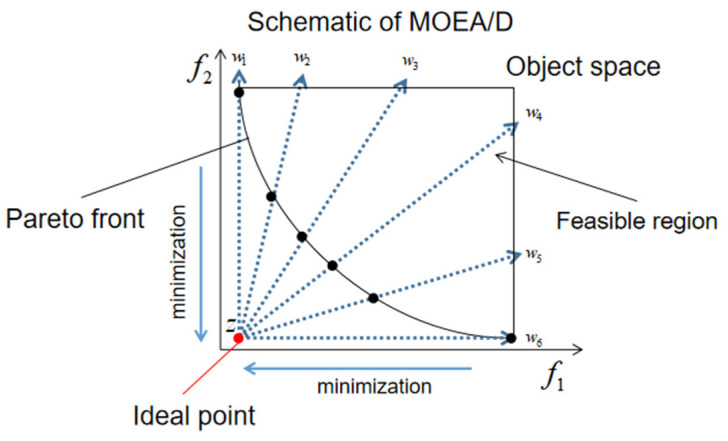
Schematic of MOEA/D. Uniformly generated weight vectors w decompose one MOP into several sub-problems.

**Figure 2 biomimetics-08-00521-f002:**
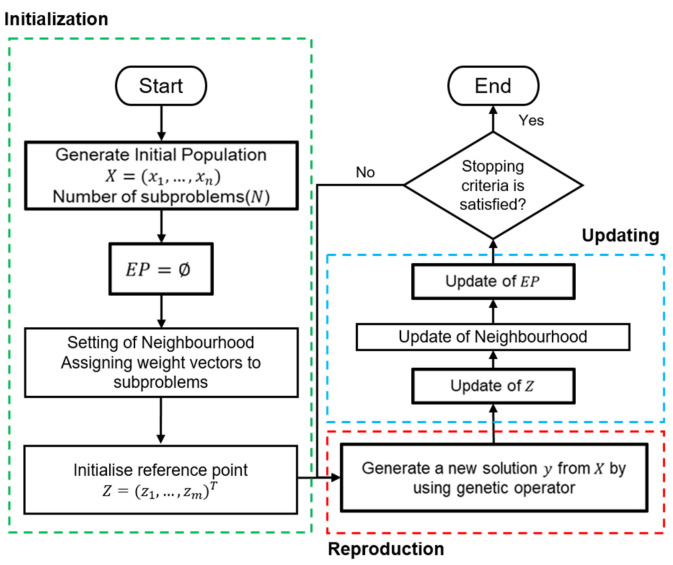
Flowchart of MOEA/D framework.

**Figure 3 biomimetics-08-00521-f003:**
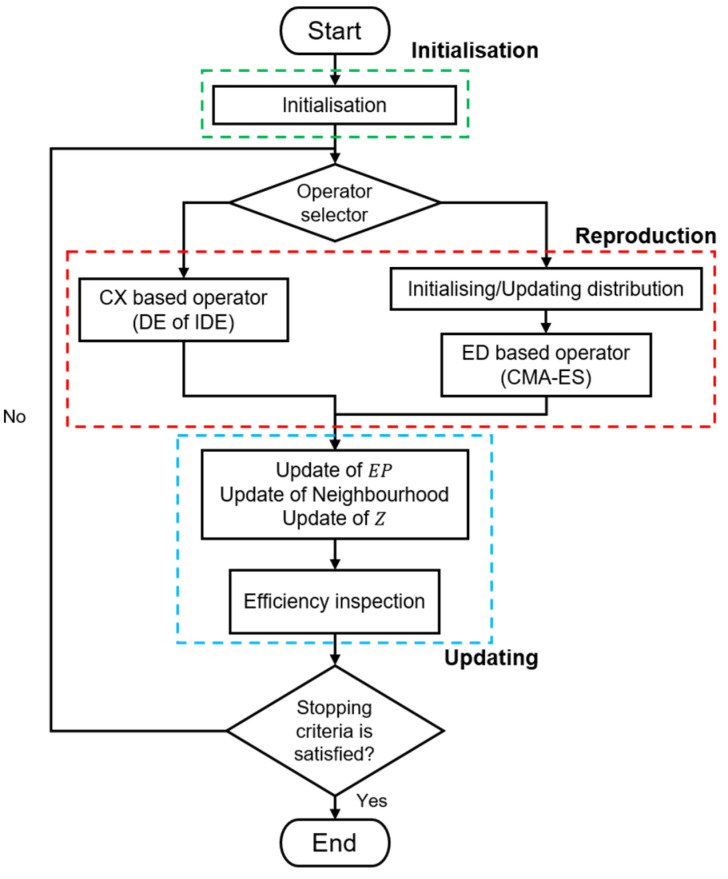
Core process of MOEA/D-HH.

**Figure 4 biomimetics-08-00521-f004:**
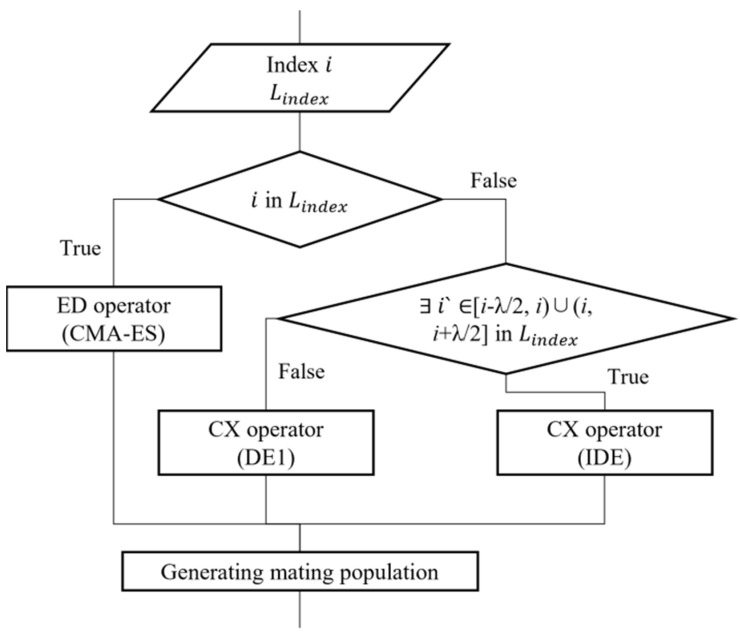
Operator selection strategy in MOEA/D-HH.

**Figure 5 biomimetics-08-00521-f005:**
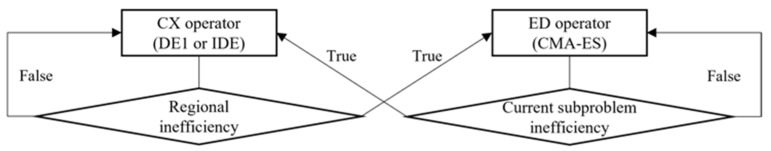
Efficiency inspection concept in MOEA/D-HH.

**Figure 6 biomimetics-08-00521-f006:**
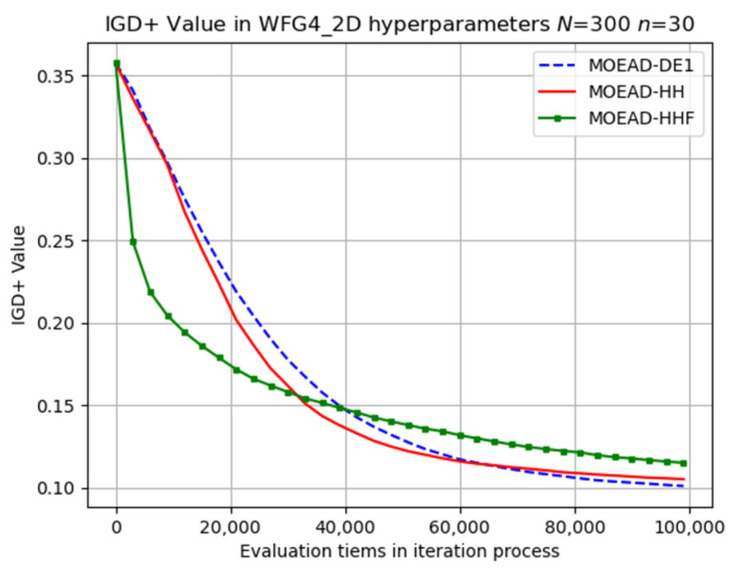
Visual evolutionary trajectory of two-objective WFG4 problem with 300 subproblems and 30 variables.

**Figure 7 biomimetics-08-00521-f007:**
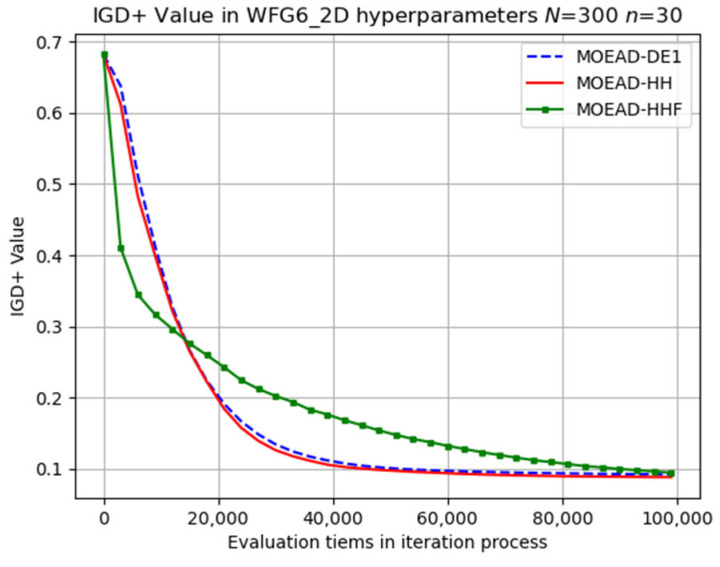
Visual evolutionary trajectory of two-objective WFG6 problem with 300 subproblems and 30 variables.

**Figure 8 biomimetics-08-00521-f008:**
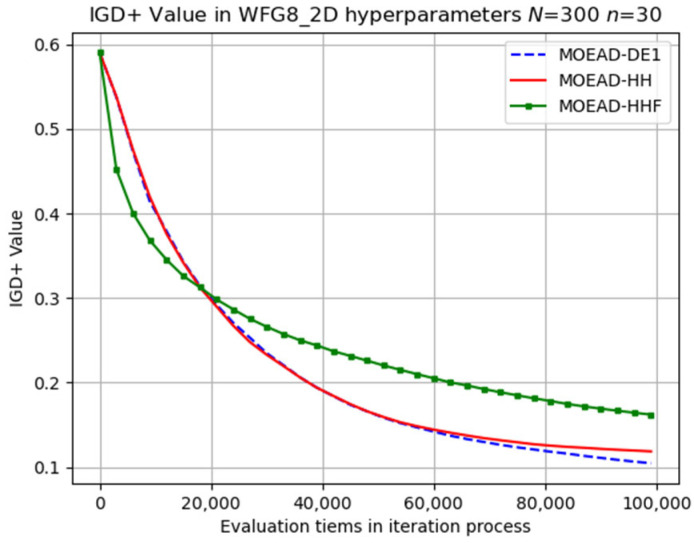
Visual evolutionary trajectory of two-objective WFG8 problem with 300 subproblems and 30 variables.

**Figure 9 biomimetics-08-00521-f009:**
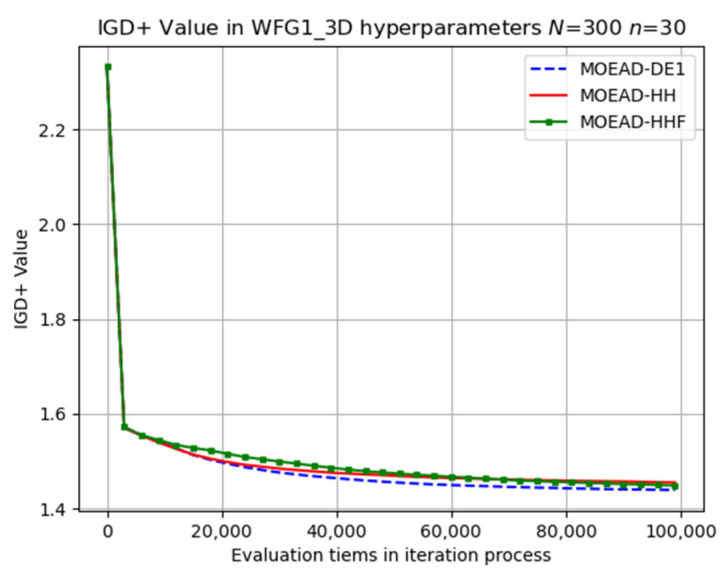
Visual evolutionary trajectory of three-objective WFG1 problem with 300 subproblems and 30 variables.

**Figure 10 biomimetics-08-00521-f010:**
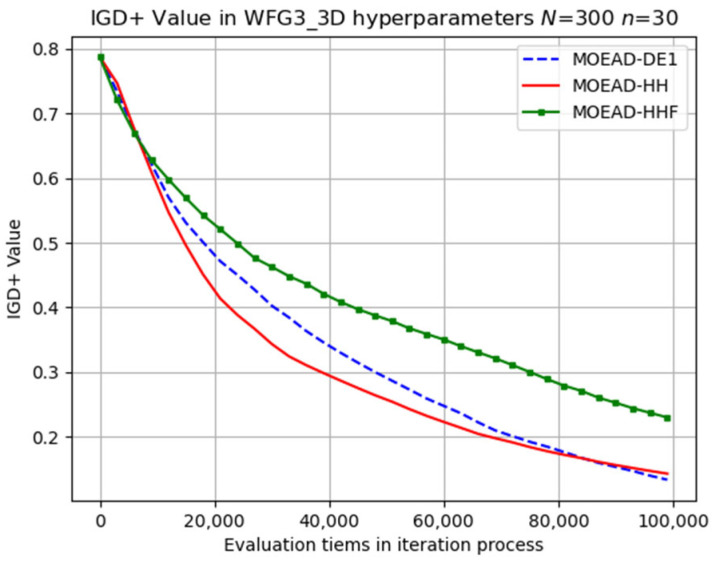
Visual evolutionary trajectory of three-objective WFG3 problem with 300 subproblems and 30 variables.

**Figure 11 biomimetics-08-00521-f011:**
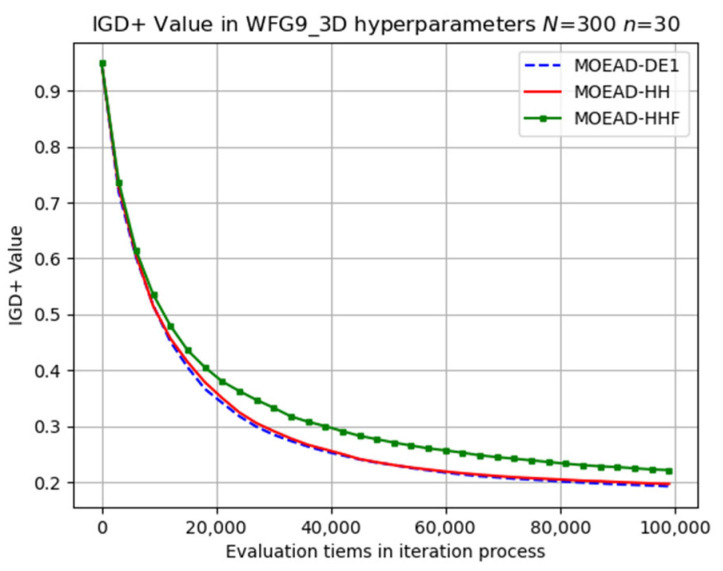
Visual evolutionary trajectory of three-objective WFG9 problem with 300 subproblems and 30 variables.

**Figure 12 biomimetics-08-00521-f012:**
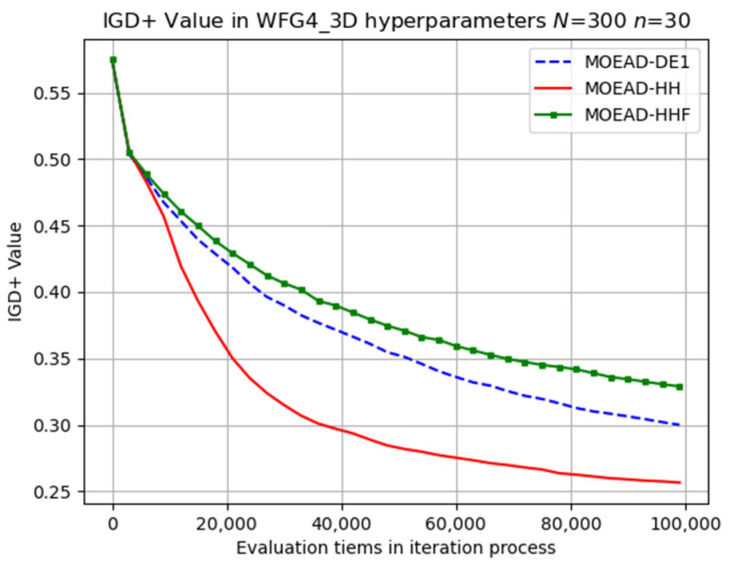
Visual evolutionary trajectory of three-objective WFG4 problem with 300 subproblems and 30 variables.

**Figure 13 biomimetics-08-00521-f013:**
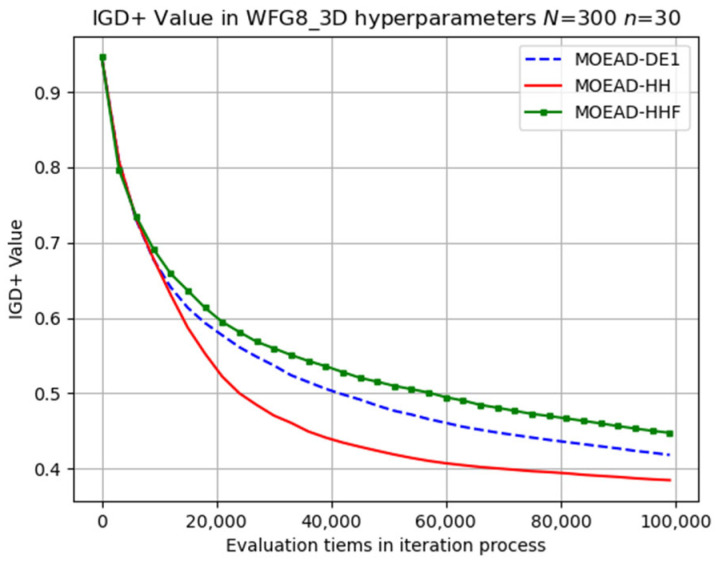
Visual evolutionary trajectory of three-objective WFG8 problem with 300 subproblems and 30 variables.

**Figure 14 biomimetics-08-00521-f014:**
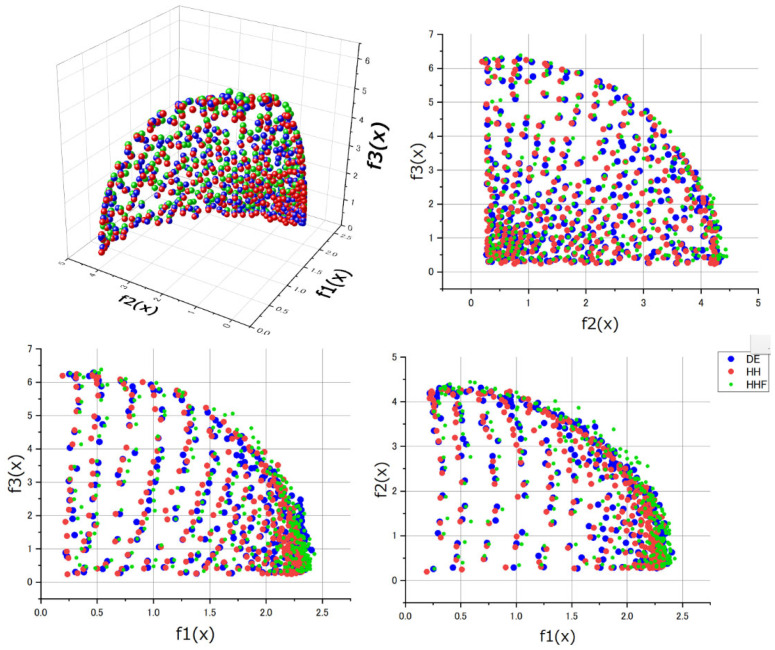
Comparison of evolutionary results for the WFG8 problem, including an overall 3D performance graph and projections on each dimension.

**Figure 15 biomimetics-08-00521-f015:**
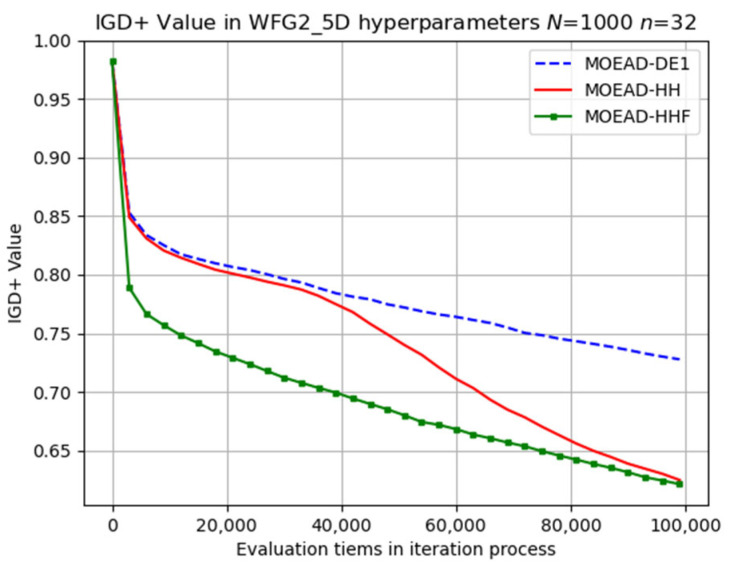
Visual evolutionary trajectory of five-objective WFG2 problem with 1000 subproblems and 32 variables.

**Figure 16 biomimetics-08-00521-f016:**
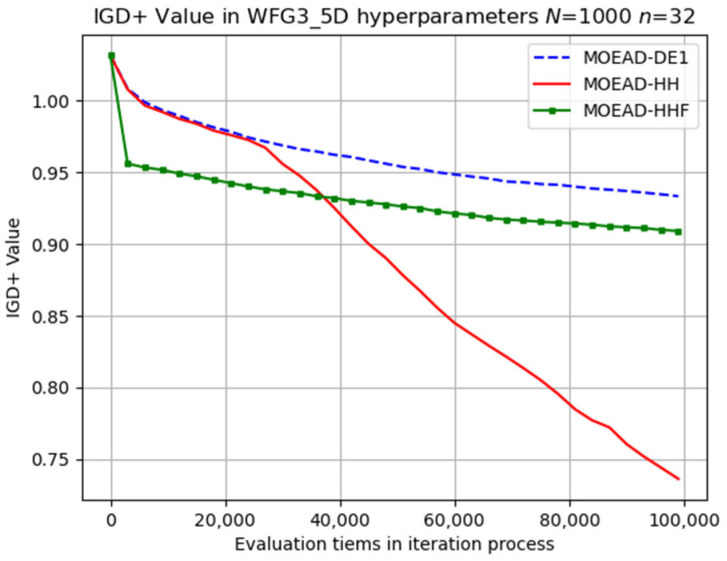
Visual evolutionary trajectory of five-objective WFG3 problem with 1000 subproblems and 32 variables.

**Figure 17 biomimetics-08-00521-f017:**
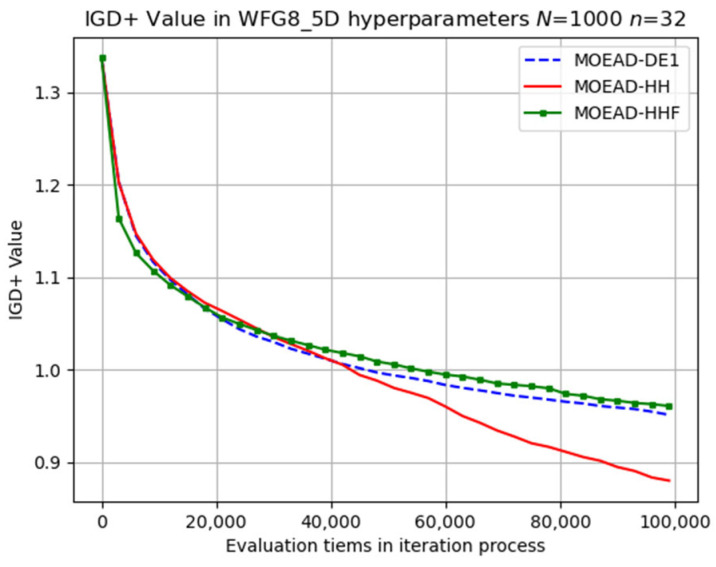
Visual evolutionary trajectory of five-objective WFG8 problem with 1000 subproblems and 32 variables.

**Table 1 biomimetics-08-00521-t001:** Average SRRs in two-, three-, and five-objective WFG problems (where # indicates the number of subsequent hyperparameter). The numerical experiment was initialised with 300 subproblems and 30 variables (1000 subproblems and 32 variables in WFG_5D problems). The iterations were repeated 21 times.

	# Objectives (M)	Two	Three	Five
Problems				
WFG1		0.1833	0.0963	0.2120
WFG2		0.2579	0.1152	0.2452
WFG3		0.2667	0.1333	0.2177
WFG4		0.1273	0.0541	0.1494
WFG5		0.1370	0.0632	0.2006
WFG6		0.1752	0.0687	0.1556
WFG7		0.2457	0.0726	0.1957
WFG8		0.2488	0.0759	0.2019
WFG9		0.1122	0.0594	0.1383

**Table 2 biomimetics-08-00521-t002:** Initialised parameters of CMA-ES in MOEA/D-HH.

Parameters	Values
m	$\mathrm{mean}\;\mathrm{value}\;\mathrm{of}\;{B(i)}^{g}$
$p_{\sigma }$	0
σ	0.5
$p_{c}$	0
C	I

**Table 3 biomimetics-08-00521-t003:** Hyper-parameter compositions in the MOEA/D framework, where indicates the number of subsequent hyperparameter.

Parameters	Values								
# Objectives (M)	2					3			5
Population size (N)	150		300			300			1000
# Design variables (n)	30	50	30	50	100	30	50	100	32
Neighbourhood size (T)	15		21			21			51
Terminal criteria (# evaluate)	100,000								

**Table 4 biomimetics-08-00521-t004:** Average value of IGD+ in WFG_2D problems.

		N=150				N=300		
		DE1	HH	HHF		DE1	HH	HHF
WFG1	n=30n=50	** 1.1343 ** ** 1.1912 **	1.2215 1.2379	1.2354 1.2489	n=30n=50n=100	** 1.1682 ** ** 1.2057 ** ** 1.2355 **	1.2107 1.2322 1.2456	1.2359 1.2463 1.2572
WFG2		** 0.0757 ** ** 0.1264 **	0.0989 0.1692	0.0966 0.1686		** 0.0608 ** ** 0.1050 ** ** 0.1800 **	0.0766 0.1260 0.1964	0.0990 0.1764 0.2437
WFG3		** 0.1356 ** ** 0.1640 **	0.1616 0.1962	0.1809 0.2535		** 0.1394 ** ** 0.1691 ** ** 0.2368 **	0.1474 0.1761 0.2379	0.1810 0.2489 0.3229
WFG4		** 0.0973 ** ** 0.1156 **	0.1068 0.1309	0.1181 0.1446		** 0.1009 ** ** 0.1207 ** ** 0.1484 **	0.1051 0.1259 0.150	0.1151 0.1442 0.1655
WFG5		** 0.0672 ** ** 0.0692 **	0.0686 0.0704	0.0729 0.0777		0.0678 0.0708 0.0752	** 0.0665 ** ** 0.0683 ** ** 0.0706 **	0.0724 0.0809 0.0990
WFG6		0.0893 0.0593	0.0889 **0.0570**	**0.0887** 0.0632		0.0924 0.0645 0.0513	** 0.0887 ** ** 0.0572 ** ** 0.0391 **	0.0949 0.0883 0.0633
WFG7		** 0.0183 ** ** 0.0356 **	0.0481 0.0728	0.0633 0.1236		** 0.0259 ** ** 0.0464 ** ** 0.0948 **	0.0365 0.0567 0.1010	0.0598 0.1112 0.1884
WFG8		** 0.1058 ** ** 0.1390 **	0.1376 0.1804	0.1799 0.2400		** 0.1046 ** ** 0.1400 ** ** 0.1959 **	0.1185 0.1499 0.2045	0.1620 0.2185 0.2757
WFG9		0.0795 **0.0558**	0.0890 0.0655	**0.0758** 0.0850		**0.0816** **0.0682** 0.0637	0.0895 0.0700 **0.0331**	0.0715 0.1000 0.1227

**Table 5 biomimetics-08-00521-t005:** SRR and PR of operators in WFG_2D problems (1).

Subproblem Number N=150		MOEA/D-DE SRR	MOEA/D-HH SRR	MOEA/D-HH		
				DE-SRR(PR)	IDE-SRR(PR)	CMA-SRR(PR)
WFG1	n=30n=50	**0.1057** **0.0921**	0.0637 0.0604	0.2928 (0.3070) 0.3003 (0.2984)	0.0513 (0.6605) 0.0501 (0.6683)	0.0344 (0.0325) 0.0328 (0.0332)
WFG2		**0.1482** **0.1579**	0.1015 0.1054	0.2830 (0.5539) 0.2883 (0.5627)	0.0405 (0.4211) 0.0455 (0.4133)	0.0541 (0.0249) 0.0718 (0.0240)
WFG3		**0.1471** **0.1580**	0.1117 0.1246	0.2923 (0.5510) 0.2828 (0.5692)	0.0489 (0.4278) 0.0707 (0.4133)	0.0251 (0.0212) 0.0723 (0.0175)
WFG4		**0.0641** **0.0710**	0.0540 0.0589	0.2269 (0.3278) 0.2306 (0.3481)	0.0510 (0.6378) 0.0563 (0.6204)	0.0437 (0.0344) 0.0516 (0.0314)
WFG5		**0.0726** **0.0785**	0.0562 0.0611	0.2619 (0.3339) 0.2587 (0.3400)	0.0424 (0.6321) 0.0514 (0.6302)	0.0044 (0.0340) 0.0071 (0.0299)
WFG6		**0.0936** **0.1015**	0.0780 0.0838	0.2895 (0.3937) 0.2895 (0.4078)	0.0512 (0.5786) 0.0567 (0.5680)	0.0028 (0.0277) 0.0025 (0.0243)
WFG7		**0.1281** **0.1378**	0.0990 0.1067	0.2789 (0.4900) 0.2686 (0.4978)	0.0581 (0.4869) 0.0765 (0.4819)	0.0479 (0.0231) 0.0830 (0.0203)
WFG8		**0.1305** **0.1357**	0.1057 0.1073	0.2869 (0.4794) 0.2779 (0.4764)	0.0682 (0.4984) 0.0790 (0.5029)	0.0685 (0.0222) 0.0900 (0.0206)
WFG9		**0.0658** **0.0737**	0.0495 0.0563	0.2283 (0.3155) 0.2316 (0.3690)	0.0432 (0.6498) 0.0466 (0.5989)	0.0211 (0.0347) 0.0246 (0.0321)

**Table 6 biomimetics-08-00521-t006:** SRR and PR of operators in WFG_2D problems (2).

Subproblem Number N=300		MOEA/D-DE SRR	MOEA/D-HH SRR	MOEA/D-HH		
				DE-SRR(PR)	IDE-SRR(PR)	CMA-SRR(PR)
WFG1	n=30n=50n=100	**0.1833** **0.1598** **0.1424**	0.1472 0.1332 0.1276	0.2907 (0.5575) 0.2938 (0.5253) 0.2846 (0.5355)	0.0937 (0.4295) 0.0893 (0.4595) 0.0772 (0.4513)	0.0533 (0.0130) 0.0553 (0.0152) 0.0447 (0.0132)
WFG2		**0.2579** **0.2664** **0.2650**	0.2225 0.2252 0.2300	0.3082 (0.8461) 0.3107 (0.8544) 0.3002 (0.8761)	0.0621 (0.1463) 0.0629 (0.1381) 0.0581 (0.1186)	0.0843 (0.0076) 0.1185 (0.0075) 0.1295 (0.0053)
WFG3		**0.2667** **0.2869** 0.2871	0.2480 0.2757 **0.2888**	0.3002 (0.9064) 0.3025 (0.9478) 0.2968 (0.9814)	0.0622 (0.0896) 0.1109 (0.0503) 0.1671 (0.0180)	0.0198 (0.0039) 0.1076 (0.0020) 0.1991 (0.0006)
WFG4		**0.1273** **0.1351** **0.1445**	0.1129 0.1196 0.1317	0.2276 (0.5780) 0.2227 (0.6084) 0.2181 (0.6522)	0.0829 (0.4054) 0.0924 (0.3773) 0.0958 (0.3379)	0.0749 (0.0166) 0.0863 (0.0143) 0.1076 (0.0099)
WFG5		**0.1370** **0.1469** **0.1613**	0.1148 0.1294 0.1420	0.2592 (0.5816) 0.2644 (0.6132) 0.2516 (0.6487)	0.0617 (0.4010) 0.0718 (0.3727) 0.0783 (0.3422)	0.0033 (0.0173) 0.0062 (0.0141) 0.0058 (0.0091)
WFG6		**0.1752** **0.2032** **0.2268**	0.1509 0.1905 0.2179	0.2936 (0.6324) 0.3005 (0.7172) 0.2947 (0.7798)	0.0744 (0.3543) 0.0964 (0.2745) 0.1105 (0.2158)	0.0025 (0.0133) 0.0054 (0.0083) 0.0122 (0.0044)
WFG7		**0.2457** **0.2625** **0.2652**	0.2265 0.2473 0.2567	0.2920 (0.8624) 0.2875 (0.9039) 0.2782 (0.9305)	0.0771 (0.1323) 0.1216 (0.0928) 0.1625 (0.0678)	0.0358 (0.0053) 0.1203 (0.0033) 0.2050 (0.0017)
WFG8		**0.2488** **0.2539** **0.2531**	0.2316 0.2413 0.2448	0.2981 (0.8415) 0.2930 (0.8660) 0.2838 (0.8845)	0.1028 (0.1532) 0.1259 (0.1299) 0.1395 (0.1127)	0.0870 (0.0054) 0.1262 (0.0041) 0.1663 (0.0028)
WFG9		**0.1122** **0.1361** **0.1649**	0.0930 0.1062 0.1315	0.2168 (0.5046) 0.2259 (0.5521) 0.2294 (0.6444)	0.0739 (0.4760) 0.0817 (0.4312) 0.0844 (0.3459)	0.0525 (0.0194) 0.0642 (0.0167) 0.0427 (0.0098)

**Table 7 biomimetics-08-00521-t007:** Average value of IGD+ in WFG_3D problems.

		DE1	HH	HHF
WFG1	n=30n=50n=100	** 1.4392 ** ** 1.4459 ** ** 1.4516 **	1.4551 1.4610 1.4632	1.4489 1.4705 1.4789
WFG2		0.3499 0.4195 0.5159	** 0.3114 ** ** 0.3618 ** ** 0.4161 **	0.3875 0.4525 0.5376
WFG3		**0.1334** 0.2580 0.3920	0.1429 **0.2390** **0.3491**	0.2299 0.3126 0.3988
WFG4		0.3000 0.3426 0.3839	** 0.2566 ** ** 0.2779 ** ** 0.2904 **	0.3289 0.3143 0.3151
WFG5		0.1728 0.1979 0.2323	** 0.1521 ** ** 0.1604 ** ** 0.1701 **	0.1866 0.2326 0.2698
WFG6		0.1913 0.1996 0.2367	** 0.1792 ** ** 0.1421 ** ** 0.1228 **	0.2102 0.2429 0.2404
WFG7		0.3225 0.4050 0.4842	** 0.2580 ** ** 0.3072 ** ** 0.3503 **	0.3462 0.3983 0.4083
WFG8		0.4180 0.4714 0.5202	** 0.3846 ** ** 0.4126 ** ** 0.4227 **	0.4474 0.4801 0.4671
WFG9		**0.1925** 0.2212 0.2651	0.1968 **0.1809** **0.1567**	0.2214 0.2567 0.2814

**Table 8 biomimetics-08-00521-t008:** SRR and PR of operators in WFG_3D problems.

		MOEA/D-DE SRR	MOEA/D-HH SRR	MOEA/D-HH		
				DE-SRR(PR)	IDE-SRR(PR)	CMA-SRR(PR)
WFG1	n=30n=50n=100	**0.0963** **0.0983** **0.0978**	0.0842 0.0850 0.0861	0.2664 (0.4828) 0.2667 (0.4955) 0.2587 (0.5298)	0.0512 (0.4892) 0.0530 (0.4775) 0.0443 (0.4475)	0.0283 (0.0281) 0.0297 (0.0270) 0.0247 (0.0227)
WFG2		**0.1152** **0.1143** **0.1089**	0.0998 0.0977 0.1000	0.2398 (0.5386) 0.2378 (0.5321) 0.2326 (0.5515)	0.0752 (0.4384) 0.0786 (0.4453) 0.0744 (0.4302)	0.0888 (0.0230) 0.0933 (0.0225) 0.0948 (0.0182)
WFG3		**0.1333** **0.1267** 0.1190	0.1173 0.1211 **0.1240**	0.2357 (0.5140) 0.2574 (0.5221) 0.2503 (0.5368)	0.1084 (0.4631) 0.1185 (0.4565) 0.1163 (0.4462)	0.1527 (0.0229) 0.1764 (0.0214) 0.1806 (0.0170)
WFG4		**0.0541** **0.0534** **0.0533**	0.0497 0.0506 0.0522	0.2363 (0.3399) 0.2362 (0.3492) 0.2244 (0.3690)	0.0455 (0.6184) 0.0484 (0.6108) 0.0480 (0.5980)	0.0481 (0.0418) 0.0481 (0.0400) 0.0529 (0.0330)
WFG5		**0.0632** **0.0639** **0.0635**	0.0567 0.0586 0.0587	0.2698 (0.3327) 0.2679 (0.3441) 0.2510 (0.3524)	0.0468 (0.6307) 0.0508 (0.6214) 0.0501 (0.6194)	0.0104 (0.0367) 0.0104 (0.0345) 0.0112 (0.0282)
WFG6		**0.0687** **0.0735** 0.0785	0.0670 0.0731 **0.0791**	0.2434 (0.3909) 0.2349 (0.4199) 0.2140 (0.4489)	0.0630 (0.5752) 0.0736 (0.5501) 0.0812 (0.5286)	0.0552 (0.0340) 0.0614 (0.0300) 0.0752 (0.0225)
WFG7		**0.0726** 0.0713 0.0666	0.0718 **0.0714** **0.0703**	0.2502 (0.3962) 0.2501 (0.4100) 0.2342 (0.4292)	0.0697 (0.5849) 0.0711 (0.5516) 0.0667 (0.5389)	0.1573 (0.0389) 0.1592 (0.0384) 0.1564 (0.0319)
WFG8		**0.0759** 0.0716 0.0680	0.0755 **0.0720** **0.0717**	0.2600 (0.4101) 0.2562 (0.4195) 0.2424 (0.4397)	0.0693 (0.5517) 0.0676 (0.5419) 0.0639 (0.5281)	0.1508 (0.0382) 0.1516 (0.0386) 0.1543 (0.0323)
WFG9		**0.0594** **0.0650** **0.0719**	0.0549 0.0594 0.0662	0.2044 (0.3834) 0.2051 (0.3815) 0.1927 (0.4206)	0.0540 (0.5818) 0.0648 (0.5868) 0.0687 (0.5561)	0.0380 (0.0348) 0.0536 (0.0317) 0.0587 (0.0233)

**Table 9 biomimetics-08-00521-t009:** Average value of IGD+ in WFG_5D problems.

	DE1	HH	HHF
WFG1	** 2.0336 **	2.0346	2.0403
WFG2	0.7278	0.6249	** 0.6214 **
WFG3	0.9334	** 0.7364 **	0.9091
WFG4	0.7183	** 0.6825 **	0.7058
WFG5	0.6621	0.6579	** 0.6431 **
WFG6	1.1639	1.1193	** 1.1156 **
WFG7	0.8753	** 0.7354 **	0.8351
WFG8	0.9514	** 0.8803 **	0.9610
WFG9	0.3865	** 0.3614 **	0.4929

**Table 10 biomimetics-08-00521-t010:** SRR and PR of operators in WFG_5D problems.

	MOEA/D-DE SRR	MOEA/D-HH SRR	MOEA/D-HH		
			DE-SRR(PR)	IDE-SRR(PR)	CMA-SRR(PR)
WFG1	**0.2122**	0.1916	0.2535 (0.8602)	0.0955 (0.1323)	0.1257 (0.0074)
WFG2	0.2342	**0.3019**	0.3134 (0.9508)	0.4769 (0.0465)	0.3647 (0.0027)
WFG3	0.2178	**0.2218**	0.3580 (0.7831)	0.1150 (0.2036)	0.0996 (0.0133)
WFG4	**0.1494**	0.1402	0.2636 (0.7645)	0.0506 (0.2185)	0.0134 (0.0170)
WFG5	**0.2006**	0.1893	0.3727 (0.7586)	0.0458 (0.2236)	0.0011 (0.0178)
WFG6	0.1556	**0.1616**	0.2765 (0.7340)	0.1306 (0.2490)	0.0987 (0.0167)
WFG7	0.1957	**0.2117**	0.3690 (0.7769)	0.1254 (0.1986)	0.3421 (0.0245)
WFG8	0.2019	**0.2043**	0.3731 (0.7672)	0.1024 (0.2075)	0.3159 (0.0253)
WFG9	0.1383	**0.1521**	0.2142 (0.7350)	0.1661 (0.2535)	0.1211 (0.0115)

## Data Availability

Data supporting reported results are available from the authors upon reasonable request.

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
