# Peer review of "A New Hyper-Heuristic Multi-Objective Optimisation Approach Based on MOEA/D Framework"

_biomimetics, 2023, doi:10.3390/biomimetics8070521_

Round 1

Reviewer 1 Report

Comments and Suggestions for Authors

This is an interesting paper that proposed multi-objective evolutionary algorithms with multiple concepted solution generators and their adaptive switching mechanism. I am not sure that this kind of approach is a part of hyper-heuristics. It is like an adaptive algorithm. This algorithm focuses on the solution update ratio switches DE, IDE, and CMS-ES adaptively. This is a well-written paper that has an impact on this research domain and should be published. My comments are as follows. The modifications are not mandatory and depend on the authors' decisions.

  • Abstract: Target issues, their reasons, and approaches to them should be described shortly. Their detail were described in the introduction. The current description only has the procedure of the proposed algorithm.
  • Abstract: The result must be clarified more. Was the target issue solved?
  • 1st paragraph of Sec. 1: CX should be explained. It would be a crossover.
  • Page 3: EX -> ED
  • Definitions of multi-objective optimization, dominance, non-dominated, etc., must be described after session 1. Also, check the variable notation with the definition. For example, X and x are used for solution notation.
  • Also, MOEA/D should be explained after that. Sec. 3.1 includes a short explanation of MOEA/D. However, Sec. 3 is the section of the proposal. MOEA/D should be explained outside of the Sec. 3. Readers cannot understand subproblems, reference point Z, etc., without any explanation.
  • Algorithm 1: "Y← ?_i = ?" would be "Y← Y \cup ?_i = ?".
  • Page 8: "thresholds for inefficiency were set were set to five and three generations for the CX and ED strategies, respectively. " It is recommended to clarify the use of these values.
  • Table 3: For the weight vector set, it is difficult to set N= 300, 1000 for 3 and 5 objectives, respectively, if the conventional simplex-lattice design is used for weight vector generation. It is recommended to clarify how the authors set weight vectors.
  • As the authors described in the conclusions, it is important to compare with other latest algorithms in the FUTURE RESEARCH, which is not this paper.

Author Response

We wish to express our appreciation to the reviewers for your insightful comments on our paper. 
The comments have helped us significantly improve the paper.

We have been able to incorporate changes to reflect most of the suggestions provided by the reviewers. The attached file presents a point-by-point response to the reviewers' comments and concerns.

Reviewer 2 Report

Comments and Suggestions for Authors

The paper presents an innovative hyper-heuristic approach integrating estimation of distribution (ED) and crossover (CX) strategies into the MOEA/D framework. This method enhances the efficiency of the DE operator by reallocating evaluation costs to the CMA-ES and IDE operators, traditionally employed for single-objective optimization problems (SOPs) despite their higher evaluation expenses compared to DE. Numerical experiments confirm a substantial enhancement in DE operator efficiency within this approach. The study is well-structured and supported by ample data, and it could be accepted after addressing the following points.

a. Has existing literature verified the applicability of single-objective optimization methods in this multi-objective optimization context? Are there previous studies demonstrating the advantages of this approach?

b. Is it effective to individually examine the effectiveness and efficacy of MOEA/D-HH by conducting experiments using the EFG test suite and MOEA/D-HHF?

c. Consider showing the Pareto Front Figure, providing an intuitive understanding of the algorithm's tolerance.

d. Is the neighborhood size freely adjustable within a specific range, or must it adhere to certain conditional formulas?

e. Instead of describing it as "simple but practical," use a more objective and specific description.

f. Regarding the statement "the problem mentioned at the beginning of this paragraph can be overcome to some extent by using MOEA/D-HH," it is recommended to verify the types of problems that can be solved by applying MOEA/D-HH, as well as the degree of improvement achieved.

g. Given the recent prominence of multiphysical problems (Mater. Horiz. 2023,10, 75-87; ACS Appl. Mater. Interfaces 2023,15,7,9940–9952), could this approach be applied to such problems? What is the applicability of this approach in the context of multiphysical problems?

Comments on the Quality of English Language

some sentence should be polished

Author Response

(The authors gave the same response as above.)

Round 2

Reviewer 2 Report

Comments and Suggestions for Authors

The revised version can be considered.

Comments on the Quality of English Language

fine